# Is the Bike Segment of Modern Olympic Triathlon More a Transition towards Running in Males than It Is in Females?

**DOI:** 10.3390/sports7040076

**Published:** 2019-03-29

**Authors:** Maria Francesca Piacentini, Luca A Bianchini, Carlo Minganti, Marco Sias, Andrea Di Castro, Veronica Vleck

**Affiliations:** 1Department of Movement, Human and Health Sciences, University of Rome “Foro Italico”, 00135 Rome, Italy; luca.bianchini.tri@gmail.com (L.A.B.); carlo.minganti@uniroma4.it (C.M.); 2Department of Biomedical Sciences for Health, University of Milan, 20133 Milan, Italy; marco.sias92@gmail.com; 3Institute of Sports Science, 00100 Rome, Italy; dicastro.training@gmail.com; 4CIPER, Faculdade de Motricidade Humana, University of Lisbon, 1499-002 Lisbon, Portugal; vvleck@fmh.ulisboa.pt

**Keywords:** endurance, elite athletes, performance

## Abstract

In 2009, the International Triathlon Union created a new triathlon race format: The World Triathlon Series (WTS), for which only athletes with a top 100 world ranking are eligible. Therefore, the purpose of this study was to analyze the influence of the three disciplines on performance within all the WTS Olympic distance races within two Olympic cycles, and to determine whether their relative contribution changed over the years. Methods: For each of a total of 44 races, final race time and position as well as split times (and positions), and summed time (and position) at each point of the race were collected and included in the analysis. Athletes were divided into 4 groups according to their final race placing (G1: 1st–3rd place; G2: 4–8th place; G3: 8–16th place and G4: ≥17th place). Two-way multivariate ANOVAs were conducted to compare the main effects of years and rank groups. For females, there were significant differences in the swim and bike segment only between G4 and the other groups (*p* range from 0.001–0.029), whilst for the run segment each group differed significantly from each other (*p* < 0.001). For males, there were significant differences in swim only between G4 and the other groups (*p* range from 0.001–0.039), whilst for the running segment each group differed significantly from the others (*p* < 0.001). Although we found running to be the segment where there were significant differences between performance groups, it is apparently important for overall success that a good runner be positioned with the first cycling pack. However, bike splits were not different between either of the four male groups or between the first 3 groups of the females. At this very high level of performance, at least in the males, the bike leg seems to be a smooth transition towards running.

## 1. Introduction

Triathlon is an endurance sport consisting of sequential swimming, swimming to cycling (T1), cycling, cycling to running (T2) and running over a variety of distances [1] and has evolved considerably since it became an Olympic sport [2]. The draft legal 1.5 km swim, 40 km bike, 10 km run event debuted at the Sydney 2000 Games and since then research has focused on the effects of one discipline on the other rather than on the effects of drafting in swimming and biking (for reviews see [3,4]).

In 2009 the International Triathlon Union (ITU) changed the racing format from a single world championship race to a series of events called the World Triathlon Series (WTS) [5] during which athletes compete head to head to collect points to become world champion. WTS is restricted to the best athletes in the world (i.e., those who are ranked up to about 150 in the ITU list, according to points that can be obtained by continental, World Cup or WTS level races). The WTS doubles as an opportunity for Olympic qualification, and athletes need to perform consistently to score the points that are necessary to be eligible for selection. Points are attributed according to race performance within a cut-off time of 5% of the best male and 8% of the best female finisher (ITU regulation-www.triathlon.org), and the World Champion is the athlete that accumulates more points and thus performs more consistently throughout the season. Seventy seven percent of races are Olympic distance and the remainder are sprint distance (involving a 0.75 km swim, a 20 km draft legal bike leg and a 5 km run). The ability of an athlete to attain ranking points depends on his/her absolute performance level (both overall and within each single discipline) and on how s/he experiences residual fatigue from the previous discipline [6]. Drafting during swimming has been shown to reduce the energy cost of a subsequent cycling bout [7,8] whilst drafting during cycling can reduce energy cost by 39% with a consequent improvement in running performance [1].

Although total race time for the Olympic distance event varies between 106–110 minutes for elite males and between 119–121 minutes for elite female athletes, it is difficult to compare one race with another because there is neither any official standardization of event distance, nor any method of weighting course difficulty (in terms of topography, climate, environmental and other factors such as drafting) in place, and all of said factors may affect the overall finishing time. Elite athletes spend about 15% of their total race time swimming, 55% of total race time cycling, and 30% of time running. Males take 17, 57 to 60 and 30 minutes respectively to do so, and females 19, 63 to 67, and 35 minutes, respectively [9,10].

Another aspect that has been investigated is the impact of each discipline on overall performance. Landers et al [11] analyzed world cup races in 1999 and reported that exiting the water in the first pack of swimmers can determine the final finishing position. In fact, 90% of male and 70% of female winners were all placed within said first pack of swimmers. Vleck et al. [12] found that, when comparing the top 50% and bottom 50% finishers of an ITU World Cup, the top 50% was faster up to the first buoy of the swim, and that thereafter the two groups did not differ in swimming speed. Moreover, overall race performance was significantly correlated with both average swimming velocity and with position after the swim stage. The speed in the bike section was not different between the top and bottom 50% of finishers but was significantly higher for the second group (lower 50%), who exited the water with the task of reducing the time gap of the leaders by the start of the triathlon run. The need to play catch up impacted negatively on athletes’ running speed. Cycling speed during the first section (13.4 km in this case) was inversely related to later running speed and the best runners were the best athletes overall. As the best triathletes can use less than 8 seconds to transition from one discipline to the other, it is also important for tactical reasons (such as avoiding collisions in the transition area) for an athlete to arrive in T2 at the front of the group [4]. Figueiredo et al. [10] reported that, in both sexes, cycling and running made the greatest contribution to the overall Olympic distance performance of top 50 finishers (i.e., approximately 36% and 47%, respectively) over the 1989–2014 period. Across the years, swimming contribution significantly decreased for women and men, whereas that of running only increased in men. However, this analysis did not take the introduction of draft legal cycling into account. Fröhlich et al. [13] showed, in males, that at Olympic distance World Championship level, running performance consistently makes the greatest contribution to which athlete wins. They concluded that “the swim and the cycle act as so-called feeders for the run and have to lay the foundations for the run, which decides over winning or losing more than the other two disciplines.” However, they performed their analysis on only one race per year. Moreover, as separate swim, T1, bike, T2 and run times were not available for all races, they used (T1 times plus swim times), and (T2 times plus bike times) together. Millet and Vleck [4] have demonstrated, in males, that individual athlete’s offset from the fastest T2 time can have an important influence on how far they eventually finish behind the race winner.

No longitudinal analyses within this genre have yet been published on all draft-legal, Olympic distance, higher level, WTS format. The purpose of the present study, therefore, was to analyze the trends in both overall and discipline specific Olympic distance triathlon performance, within the WTS only, of different performance levels of male and female triathletes, for the 2009–2016 period. The second aim was to study the differential times (i.e., the time differences to the fastest split time at that moment of the race) in T1 or out of T2. We hypothesized that swimming and cycling performance levels would level off over time and that running would be the main distinguishing factor between medalists and non-medalists. Our secondary hypothesis was that the entry into T1 would be less important than the exit from T2 in male athletes compared to female athletes. Because females seem less able than males at bridging cycle packs [6], the entry into T1 would be a determinant aspect of overall performance.

## 2. Materials and Methods

The data for this study were retrieved from the ITU world triathlon series website (wts.triathlon.org) and took into consideration only the WTS Olympic distance races from 2009–2016 for both sexes, including the most two recent Olympic Games (London 2012 and Rio 2016). Because the data are public and available on the internet, no formal ethics committee approval was necessary. For each race, final race time and position as well as split times (and positions) and summed time (and position) at each point of the race (S, S + T1, S + T1 + B, S + T1 + B + T2, S + T1 + B + T2 + R, where S, B and R equate to swimming, cycling and running, respectively) were retrieved and included in the analysis. In total, 44 races and 1670 male and 1706 female performances were examined (Table 1).

Thereafter, for each race, participants were divided into 4 groups according to their final race placing i.e., G1: 1st–3rd place; G2: 4–8th place; G3: 8–16th place and G4: ≥17th place. Those athletes who did not finish the race in question were excluded from analysis. In accordance with the results of previous work that has highlighted the importance of the exit from the swim and the positioning in running to final race performance [3,6,12], the raw times were converted into differential times (offset from 1st in that leg) for T1 and T2 only.

### Statistical Analysis

The statistical package IBM SPSS version 20 (IBM, Chicago, IL, USA) was used for the analysis. Values are presented as mean and standard deviations and before the analysis, the Kolmogorov–Smirnov test was applied to test the normal distribution of the data. The men’s and the women’s data were treated separately. Two-way multivariate ANOVAs were conducted to compare the main effects of years (i.e., from 2009–2016) and rank groups (i.e G1, G2, G3, G4) on time measures for each component of the race (i.e the swim, bike, and run). Before the analysis, the Levene’s test for homogeneity of variance was performed to verify the assumption of the test. Furthermore, separate ANOVAs were conducted to compare the effects on differential times in T1 and T2 (offset from the 1^st^ in T1 and from the first out of T2) per year and per group. The same analysis was conducted taking only the medalists into account.

Univariate effects within MANOVAs were examined only if the overall MANOVA was significant. When significant interaction was observed (years for rank groups), follow-up tests were conducted by splitting the sample into the four rank groups and running separate ANOVAs to explore the different effect of years. When multiple comparisons were performed, post-hoc Fisher’s protected least significant difference (LSD) test with Bonferroni correction was used.

The significance level for all comparisons was set at P ≤ 0.05. In addition, effects size (ES) were calculated for all variables as partial eta-squared (η2p). Partial eta-squared values below 0.01, between 0.01–0.06, between 0.06–0.14, and above 0.14 were considered to have trivial, small, medium, and large effect sizes, respectively [14]. 

## 3. Results

For female athletes, analysis showed a multivariate effect for years (Wilks’ λ = 0.867; F_21,1685_ = 11.68, *p* < 0.001; η2p = 0.047) and for rank groups (Wilks’ λ = 0.639; F_9,1697_ = 91.22, *p* < 0.001; η2p = 0.139) although no interaction effect (years for rank groups) was found.

Univariate tests indicated significant effects on the three disciplines both by year (swim: F_7,1699_ = 9.052; *p* < 0.001; η2p = 0.036 – bike: F_87,1699_ = 14.47; *p* < 0.001; η2p = 0.057–run: F_7,1699_ = 7.23; *p* < 0.001; η2p = 0.029) and by group (swim: F_3,1704_ = 9.84; *p* < 0.001; η2p = 0.017–bike: F_3,1704_ = 4.24; P = 0.005; η2p = 0,008–run: F_3,1704_ = 295.07; *p* < 0.001; η2p = 0.346). Post-hoc for rank groups showed significant differences in the swim and bike segment only between G4 and the other groups (p range from 0.001–0.029) whilst for the run segment each group differed significantly from each other. (*p* < 0.001) (Figure 1)

The significant differences in the post-hoc by year for total time and for single legs are not reported because no trend of particular interest was noticed.

The analysis of the males showed a multivariate effect for years (Wilks’ λ = 0.849; F_21,1649_ = 13.17, *p* < 0.001; η2p = 0.053) and for group (Wilks’ λ = 0.601; F_9,1661_ = 102.90, *p* < 0.001; η2p = 0.156) while no interaction effect (year for rank groups) was found. Univariate tests indicated significant effects on the three disciplines by year (swim: F_7,1663_ = 11.49; *p* < 0.001; η2p = 0,047–bike: F_7,1663_ = 18.65; *p* < 0.001; η2p = 0.074–run: F_7,1663_ = 7.19; *p* < 0.001; η2p = 0.039).

By group differences were found for swimming (F_3,1667_ = 6.21; *p* < 0.001; η2p = 0.011) and for the run (F_3,1667_ = 321.86; *p* < 0.001; η2p = 0.371) while no effect was found for the bike leg between groups. Post-hoc for rank groups (Figure 2) showed significant differences in swim only between G4 and the other groups (p range from 0.001–0.039), while for the running segment each group differed significantly from the others (*p* < 0.001).

Similarly to the women, the significant differences in the post-hoc by year for total time and for single legs are not reported because no trend of particular interest was noticed.

Analysis of differential times in entry in T1 and exit from T2 (offset from the 1st in that moment of the race) for women (Table 2) showed differences by rank groups in T1 (F_3,1703_ = 16.38 *p* < 0.001, η2p = 0.040), with G1 (medalists) being significantly different from the others (*p* < 0.001). For exit from T2, all groups differed from each other (F_3,1703_ = 65.69; *p* < 0.001, η2p = 0.142). For males (Table 3), the analysis showed a difference by rank groups both in entry in T1 (F_3,1663_ = 42.01, *p* < 0.001, η2p = 0.091) and in exit from T2 (F_3,1663_ = 45.05 *p* < 0.001, η2p = 0.100), with only the first group differing (*p* < 0.001) from the others in both cases.

Finally, when analyzing only G1 (medalists), no effect per year, final position or interaction years by position were found both for entry in T1 and exit in T2. Table 4 shows the position of each athlete of G1 in entry in T1 in exit of T2 and run split, for women and men respectively for the whole period (2009–2016).

## 4. Discussion

The purpose of the present study was to analyze the contribution of each segment of the Olympic distance triathlon to overall performance since the inception of the new WTS race format. With this new format athletes gain points that count towards the allocation of the 55 start places for the Olympic Games [9] within the WTS, within which only the best athletes in the world are allowed to compete. Our main finding was that there were differences in swim time, bike time and run time both per year and per group in the females. Within the males, such differences were observed only for swimming and running. Regarding swimming, we found no difference in segment times, in both sexes, between the first 3 groups (best 16 athletes overall)

Researchers have increasingly attempted to identify the optimal strategy for triathlon success since it became an Olympic sport, via assessment of predictors of performance, of the impact of one triathlon discipline on the other as compared to its component single disciplines, and of the best strategy to adopt during a race [2]. Although recent studies have put forward different analyses to understand the impact of the three disciplines on overall performance [9,10,13,15], the current study is the first to compare the performances over two Olympic cycles of the very top-level group of elite triathletes who competed in the WTS. Indeed, we observed smaller time differences between groups and athletes at the WTS level, i.e., every athlete which was ranked within the top 150 worldwide, than have been recorded in the literature to date. This is likely because most of said studies and some of the previously reported studies [10,13] examined performance in lower tier (albeit still elite) events, which have less restrictive criteria for athlete eligibility. Due to the strict selection process, on average only 60 athletes are allowed to start each WTS event. Various studies have previously recorded very high correlations between swimming prowess and overall performance. Landers et al. [11] showed that 90% of male winners and 70% of female winners exited in the first pack of swimmers. Ofoghi et al. [16] utilized Bayesian networks to analyze the differential time, calculated as the difference between each time in that segment of the race and the first athlete at that moment of the race, at different race points and use it to predict the likelihood of finishing on the podium. They reported that the medalists swam significantly faster than the lower placed athletes (i.e., those who finished 4–10th and above 10th place). On the other hand, Cejuela et al., [17] when analyzing the top races from 2000–2008, observed a low correlation between swimming and overall performance. Therefore, it seems that at least for the best athletes (placing up to 16th overall in the event) swimming performance levelled off from the introduction of new WTS format, and that swimming performance does not discriminate between different performance levels at the top level of professional triathlon.

Nonetheless, swimming position still seems to be a good determinant of overall success [6,11,12] because, although its contribution might be low, the strategic positioning within this segment may be critical to overall race performance. In fact, just analyzing contribution of each segment (or changes per year) does not reflect the race dynamics in relation to the race leaders in each moment of the race [10]. We therefore analyzed the offset of each athlete per group from the first athlete entering T1. We found that the men in G1 were generally 16” off from the fastest swimmer, and that this offset was higher in the other performance groups (G2: 24”, G3: 29”, G4: 34”). No differences were observed between the first 3 groups in the females but not only did G4 females swim significantly slower, but the G4 female group was the group with the highest offset from the fastest swimmer.

Exiting the water with a limited offset from the fastest swimmer, was shown to influence entry into the first pack(s) in the bike leg, influencing overall finishing position [6]. Depending on the individual athletes relative cycling and running ability [18], failure to enter said pack(s), and a subsequent increase in power output/effort during the bike leg in order to try to catch up to the front pack(s), may then negatively impact running performance. In accordance with the results of previous studies [6,17] we found no differences in bike splits within male different ranking groups. As regards to the women, only the first 16 athletes (G1 to G3) had similar bike splits confirming that even at this level females are divided in more groups, further apart, as compared to males.

Therefore, running seems to have the major impact on overall WTS race performance. Figuereido et al [10] showed that, over a 26-year period, the average contribution of the cycling and running stage has much more impact on overall performance of the top 50 Olympic distance finishers compared to swimming, irrespective of athlete sex. Moreover, they found that for women, swimming contribution decreased while cycling and running contribution remained unchanged. For males, running contribution significantly increased over time. Fröhlich et al [13] analyzed individual data from the world championships from 2003–2007 for all finishers as compared to the top 20, using multiple linear regression. They again highlighted the importance of running in the Olympic distance triathlon, as the discipline where performance is most related to the overall time and finishing position. Clearly, above average running performance is essential to placing well, especially in G1. Moreover, the better cyclists need to be able to keep the best runners behind them in order to have a chance to perform optimally. All the groups that were analyzed in the present study differed for running time, being G1 the fastest. We found that, over all the years and races that were analyzed, both the female and the male winners had, on average, the 2nd run split; the second finisher exhibited, on average, the 4th run split; whilst the third finisher had, on average, the 5th run split -despite there being no particular differences between the first three athletes in their position at the exit from T2 (Table 2). This appears to confirm that running is the most important determinant of overall triathlon performance [10], and that it is crucial for an athlete to arrive in the transition (T2) at the front of the group to avoid collisions or jams [4].

The second transition has been considered another important segment of the race. Because athletes often finish within seconds of each other, run times have levelled off and exiting T2 fast is important to gain precious seconds. Interestingly, male G1 was the group that over the years came closer to the exit from T2, (from 40” in 2009 to only 9” in 2016). In contrast, G3 and G4 slowed down over the years (increasing the gap from the first athlete out of T2). This could be partly because men generally arrive in larger groups in T2, as compared to female athletes, and therefore need to exit extremely quickly from T2 in order to gain precious seconds on their opponents. For the women, all the groups differed for T2 meaning that G1 was the group that was closer in time to the race leader at that moment, with no particular differences observed over the years. This supports the assertion that females may be less able to coalesce different bike packs into one [6], and that less athletes consequently enter transition zones at the same time.

Similarly, Cejuela et al [17] studied nine top level male competitions from 2000–2008. Their analysis included the lost time in each transition—calculated as the time lag between the first triathlete who started cycling or running and the rest of the athletes who arrived in the transition area with the same pack. They found a low correlation between T1 and overall performance; however, the lost time in T1 was different for each swimming pack (when 5” gaps between swimmers were taken to indicate a different pack). Even in this case, they found very low correlations between time lost and overall performance, probably because of the very flat biking routes that allowed groups to reunify. On the other hand, they confirmed that time lost in T2 is inversely related to performance. Considering the levelling off of running performance, the quicker the exit from T2 (and lower time loss) will for sure be beneficial for overall performance and final positioning.

Although, as already reported, we found running to be the segment where there were significant differences between performance groups, [17], it is apparently important for overall success that a good runner be positioned with the first cycling pack in order to have the possibility to win the race [12]. However, bike splits did not differ between male groups. At this very high level, the bike leg seems to be a smooth transition towards running, at least for the male athletes. 

In conclusion, for males it appears that exiting the water close to the first athlete and exiting T2 close to the first athlete, with a fast running split, is a major determinant of success. For the women, exit from both T1 and T2 seem a major determinant of performance, as is a very fast running split. Over the years, the offset of G1 from the first athlete to exit T2 remained stable, whilst that of G2–G4 significantly worsened. The gender difference we observed in the relative influence of performance within specific sections of WTS competition to overall result can be explained by the greater number of bike packs that are seen in the women’s races and their different race tactics. Because females seem less able than males at bridging cycle packs [6], their entry into T1, in contrast to males, seems to be a key aspect of overall performance.

### Practical Applications

Based on the results of this analysis, we would suggest that both for males and females it is worthwhile to train the actual practice of T2 transitions. This is particularly the case in those athletes who are weaker overall, and- who will consequently (as this is a function of ranking) have their bikes placed further from the transition exit, and therefore likely have more potential to be caught up in "traffic jams." In line with the findings of Vleck et al. [6], strengthening of female biking ability to the point that athletes become better able to bridge gaps to a leading pack (through the ability to sustain short high power output bursts immediately followed by steady lower level effort, and improvement of climbing ability in the case of hilly courses) is advisable, as, depending on the athlete, it may have a significant influence on overall race placing. Run training (“brick” sessions) and performance remain of paramount and increasing importance but at present the disciplines that precede the triathlon run appear to have more impact on overall race performance in females than they do in males. The pacing characteristics of performance at this level have not as yet been established but this can clearly be the key to overall race placing, particularly in males where the performance density is better and the ability to complete a fast, sprint type, run finish can be definitive. Moreover, the data presented in this paper may prove be helpful in the selection of young talented athletes, who need to compete at a young age and not come into triathlon from one of its disciplines in isolation. 

## Figures and Tables

**Figure 1 sports-07-00076-f001:**
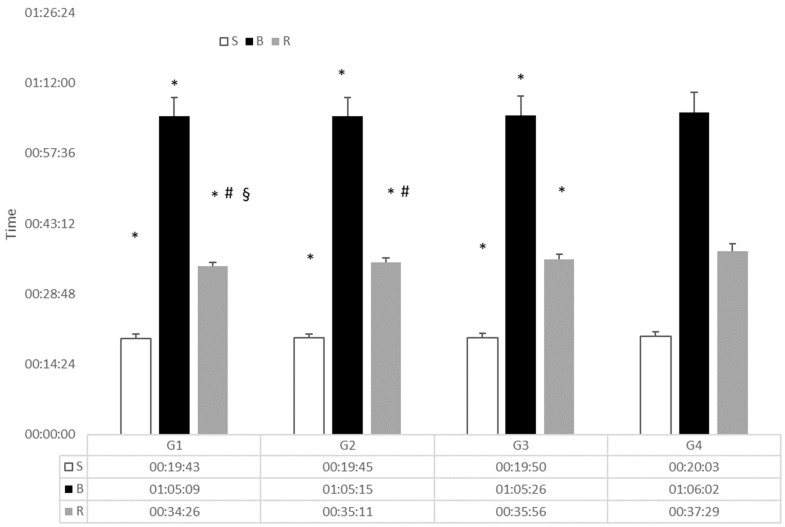
Split segment times for each of the female groups. S: Swim, B: bike, R: run, * different from G4, # diff from G3, § diff from G2.

**Figure 2 sports-07-00076-f002:**
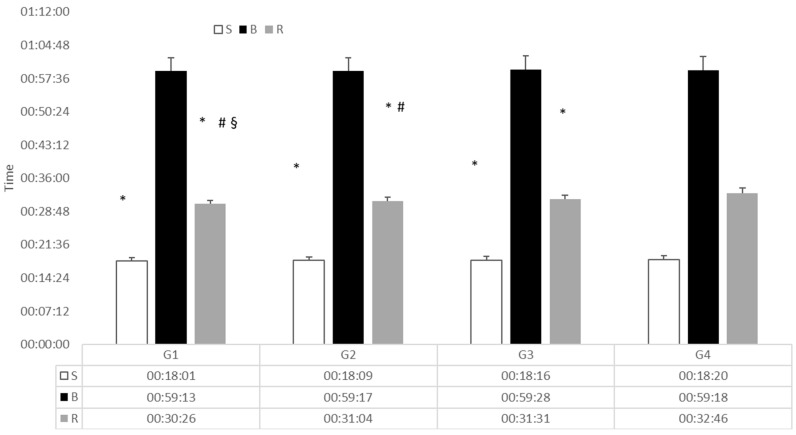
Split segment times for each of the male groups. S: Swim, B: bike, R: run, * different from G4, # diff from G3, § diff from G2.

**Table 1 sports-07-00076-t001:** Represents the number of races and athletes analyzed each year.

Year	Races	Females	Males
2009	8	239	266
2010	6	266	271
2011	6	308	286
2012	6	268	204
2013	5	151	142
2014	4	113	150
2015	4	163	156
2016	5	198	195
Total	44	1706	1670

**Table 2 sports-07-00076-t002:** Offset time from the first in T1 and the first out of T2 for each group of female athletes.

Groups	Offset Entering T1 (sec)	Offset Existing T2 (sec)
G1	28 ± 12	12 ± 8
G2	30 ± 8 *	19 ± 15 *
G3	35 ± 7 *	35 ± 15 *#
G4	43 ± 10 *	66 ± 26 *#§

* different from G1, # diff from G2, § diff from G3 *p* < 0.05.

**Table 3 sports-07-00076-t003:** Offset time from the first in T1 and the first out of T2 for each group of male athletes.

Groups	Offset Entering T1 (sec)	Offset Exiting T2 (sec)
G1	15 ± 2	21 ± 11
G2	24 ± 4 *	38 ± 10 *
G3	30 ± 5 *	54 ± 16 *
G4	35 ± 6 *	61 ± 15 *

* different from G1 *p* < 0.05.

**Table 4 sports-07-00076-t004:** Position of the medalists in T1, T2 and run split over the years.

	Females			Males		
	1st	2nd	3rd	1st	2nd	3rd
T1 position	14 ± 5	16 ± 5	13 ± 4	10.2 ± 4.5	14.3 ± 4	18.2 ± 4
T2 position	8.6 ± 4	8.2 ± 4	7.4 ± 1.5	7.7 ± 4	9.6 ± 3	10.7 ± 2
Run split	2.1 ± 1	3.8 ± 1.5	5.3 ± 2.2	2.2 ± 0.8	3.8 ± 2.6	4.5 ± 1.4

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
