# Peer review of "Is the Bike Segment of Modern Olympic Triathlon More a Transition towards Running in Males than It Is in Females?"

_sports, 2019, doi:10.3390/sports7040076_

Round 1

Reviewer 1 Report

This is a very interesting study. The authors analysed the influence of the three disciplines on all WTS Olympic distance races performed on 2 Olympic cycles, and if the contribution has changed over the years. Results showed the importance of running in the current olimpic distance triathlon. The rational of this study is clear. Methodological issues are clear. However, I offer the following suggestions for improvement

-Please, review the format of the references in the text. Sometimes you use the number and sometimes the year. Please amend. 

-Please, review some references with mistakes (for example Cejuela)

-Please, explain why do you use this classification in four groups or categories. What is the rationale of this clasiffication? 

-Explain training application of the results obtained for triathlon population

Author Response

We are grateful to the Editor for giving us the possibility to revise the manuscript and making it suitable to be considered for publication. The reviewers’ comments proved exceptionally helpful in the revision process. The manuscript was revised in structure and content according to their suggestions. Each comment was carefully addressed with a specific answer (R) and the respective changes were inserted in the manuscript in red.

Reviewer 1

This is a very interesting study. The authors analysed the influence of the three disciplines on all WTS Olympic distance races performed on 2 Olympic cycles, and if the contribution has changed over the years. Results showed the importance of running in the current Olympic distance triathlon. The rational of this study is clear. Methodological issues are clear. However, I offer the following suggestions for improvement

We thank the reviewer for the positive feedback and responded point to point to his/her useful suggestions.  

-Please, review the format of the references in the text. Sometimes you use the number and sometimes the year. Please amend. 

WE thank the reviewer for pointing out this issue,. WE changed throughout the text according to the authors guidelines

-Please, review some references with mistakes (for example Cejuela)

Thank you again we realized there was a mistake in the year of publication This has been changed

-Please, explain why do you use this classification in four groups or categories. What is the rationale of this classification? 

WE used a  qualitative inclusion for medalists and finalists (considering a top 8 being finalist in most sporting events (i.e swimming, track and field) (Rodriguez et al. 2017, Baldassarre et al. 2018)

-Explain training application of the results obtained for triathlon population

We thank the reviewer for pointing this out. A specific paragraph has been added at the end of the conclusions

Author Response

We are grateful to the Editor for giving us the possibility to revise the manuscript and making it suitable to be considered for publication. We thank the reviewer for the positive feedback.  The reviewers’ comments proved exceptionally helpful in the revision process. The manuscript was revised in structure and content according to his/her suggestions and the respective changes were inserted in the manuscript in red.

As suggested we changed the title of the manuscript pointing out that we discuss differences between male and female performances. 

Moreover, as suggested, we added a specific paragraph at the end of the discussion on possible practical applications for athletes and coaches

Round 2

Reviewer 2 Report

First, I would like to thank the authors for addressing my comments. Please refer to the attached file for further comments/suggestions. 

Author Response

We thank again the reviewer for his/her feedback in improving the quality of the manuscript.

The changes have been highlited in the text